# Robust Variational Contrastive Learning for Partially View-unaligned Clustering

Changhao He
hechanghao.gm@gmail.com
Sichuan University
Chengdu, China

Hongyuan Zhu
hongyuanzhu.cn@gmail.com
Institute for Infocomm Research, A*STAR
Singapore

Peng Hu*
penghu.ml@gmail.com
Sichuan University
Chengdu, China

Xi Peng
pengx.gm@gmail.com
Sichuan University
Chengdu, China

## Abstract

Although multi-view learning has achieved remarkable progress over the past decades, most existing methods implicitly assume that all views (or modalities) are well-aligned. In practice, however, collecting fully aligned views is challenging due to complexities and discordances in time and space, resulting in the Partially View-unaligned Problem (PVP), such as audio-video asynchrony caused by network congestion. While some methods are proposed to align the unaligned views by learning view-invariant representations, almost all of them overlook specific information across different views for complementarity, limiting performance improvement. To address these problems, we propose a robust framework, dubbed **V**ariat**I**onal Con**T**r**A**stive **L**earning (VITAL), designed to learn both common and specific information simultaneously. To be specific, each data sample is first modeled as a Gaussian distribution in the latent space, where the mean estimates the most probable common information, while the variance indicates view-specific information. Second, by using variational inference, VITAL conducts intra- and inter-view contrastive learning to preserve common and specific semantics in the distribution representations, thereby achieving comprehensive perception. As a result, the common representation (mean) could be used to guide category-level realignment, while the specific representation (variance) complements sample semantic information, thereby boosting overall performance. Finally, considering the abundance of False Negative Pairs (FNPs) generated by unsupervised contrastive learning, we propose a robust loss function that seamlessly incorporates FNP rectification into the contrastive learning paradigm. Empirical evaluations on eight benchmark datasets reveal that VITAL outperforms ten state-of-the-art deep clustering baselines, demonstrating its efficacy in

both partially and fully aligned scenarios. The Code is available at https://github.com/He-Changhao/2024-MM-VITAL.

## CCS Concepts

• **Theory of computation** → **Unsupervised learning and clustering**; • **Computing methodologies** → **Cluster analysis**.

## Keywords

Partially View-unaligned Clustering, Variational Contrastive Learning, False Negative Pairs

**ACM Reference Format:**
Changhao He, Hongyuan Zhu, Peng Hu, and Xi Peng. 2024. Robust Variational Contrastive Learning for Partially View-unaligned Clustering. In *Proceedings of the 32nd ACM International Conference on Multimedia (MM '24), October 28-November 1, 2024, Melbourne, VIC, Australia.* ACM, New York, NY, USA, 10 pages. https://doi.org/10.1145/3664647.3681331

## 1 Introduction

Multi-view learning aims at exploiting the common and specific information of different views (or modalities) to achieve comprehensive perception [13, 34]. The success of existing multi-view learning methods heavily relies on an implicit assumption that all views are aligned perfectly [4, 28, 31]. In practice, however, this assumption is not always feasible, due to sensor discrepancies or communication disruptions, resulting in unaligned views, i.e., Partially View-unaligned Problem (PVP) [15, 39]. That is to say, the collected views lack proper alignment, which will adversely affect multi-view representation learning and, consequently, downstream tasks [29, 43].

Traditional solutions to PVP involve graph matching techniques, such as the Hungarian algorithm [20], to align unaligned views in a common space. However, their non-differentiable nature precludes integration with deep neural networks (DNNs) for end-to-end learning. To overcome this issue, Huang et al. [15] presented Partially View-unaligned Clustering (PVC), a differentiable alternative that learns common representations across different views while achieving differentiable alignment by using these representations. Nonetheless, PVC, akin to the Hungarian algorithm, relies on instance-level alignment, which is excessive for tasks requiring only category-level alignment [38, 39], such as clustering. Recent advancements, such as MvCLN [39], introduced category-level alignment, enhancing efficiency and reducing matching errors brought

*Corresponding author

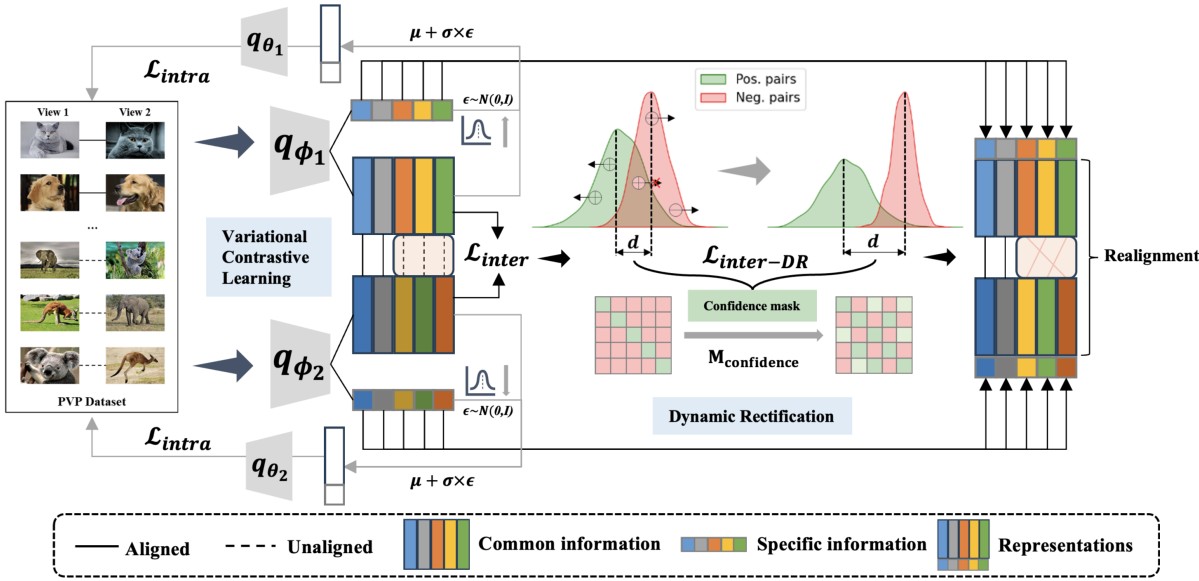

**Figure 1: Our VITAL consists of two modules: Variational Contrastive Learning and Dynamic Rectification. For variational contrastive learning, we utilize a probabilistic encoder ($q_\phi$) to approximate the true posterior distribution of observed samples, where the mean and variance of the distribution are modeled as common and specific information, respectively. Then, $\mathcal{L}_{inter}$ is used for contrasting between different views in the common space to maximize the shared semantics, while $\mathcal{L}_{intra}$ is used for contrasting between reconstructed and original views to retain intrinsic semantics. As for Dynamic Rectification, we employ a probabilistic model to fit the loss and derive a confidence mask for FNPs (see section 3.4). Finally, the common representations are utilized for realigning views, and the features ultimately used for downstream tasks are obtained from the combination of common and specific representations.**

by instance-level alignment. SURE [38] extended MvCLN by reconstructing each original view from integrated common representations, aiming to preserve high-dimensional semantic information. Yet, these methods focused predominantly on common information, often at the expense of unique view-specific information, leading to suboptimal representations for downstream applications.

To address this problem, we propose a robust framework, dubbed **V**ariat**I**onal Con**T**r**A**stive **L**earning (VITAL), to learn common representations for category-level realignment and view-specific representations for information complementation. The pipeline of VITAL is shown in Figure 1, which consists of two modules: Variational Contrastive Learning and Dynamic Rectification. Specifically, VITAL first models each data sample as a Gaussian distribution in the latent space, with the mean representing likely common information and the variance capturing specific information. Second, to preserve comprehensive semantic information, VITAL employs intra- and inter-view contrastive learning coupled with variational inference to encapsulate intrinsic semantics into the distribution representations and maximize the shared semantics across different views, respectively. Thanks to the explicit learning of both view-invariant and view-specific information, our VITAL could infuse comprehensive semantics into the latent space, thereby enhancing the discrimination of the fused representations. Finally, as the labels are unavailable, all unpaired samples across different views are considered as negative pairs, which inevitably leads to some samples from the same class being incorrectly treated as negatives,

namely False Negative Pairs (FNPs). Clearly, FNPs will mislead the models, resulting in suboptimal or even erroneous solutions. To alleviate or even eliminate the adverse impact of FNPs, we present a robust contrastive loss to adaptively focus more on clean pairs and less on noisy ones, thus avoiding overemphasizing FNPs and boosting the model's robustness. Our main contributions are as follows:

- We propose a comprehensive framework to tackle the Partially View-unaligned Problem (PVP), which initially aligns views using common representations, and subsequently enriches the fused representations with specific information.
- To model both common and specific information, we present a novel Variational Contrastive Learning paradigm to learn view-invariant representation (mean) and view-specific one (variance) simultaneously.
- A robust inter-view contrastive loss is proposed to selectively optimize clean and noisy pairs, thus alleviating or even eliminating the negative impact of FNPs during training.
- Our extensive experiments on eight benchmark datasets, encompassing both partially and fully aligned scenarios, corroborate the effectiveness and superiority of our framework.

## 2 Related Works

This section provides a brief review of some related works in the domains of multi-view learning and contrastive learning.

## 2.1 Multi-view Learning

Most existing multi-view learning methods implicitly take the view completeness assumption to learn from multiple views or modalities [13, 14, 28]. However, this ideal assumption would be easily broken in practice, resulting in incomplete views, i.e., Partially Sample-missing Problem (PSP) and Partially View-unaligned Problem (PVP). To address PSP, numerous methods were proposed to learn view-invariant representations between views through filling in the missing parts using information from complete portions [22, 24–26, 30]. Additionally, the PVP-oriented approaches aimed to establish the cross-view correspondences of the view-unaligned samples by learning view-invariant representations [15, 38]. The pioneering PVP-oriented method PVC utilized DNNs to reconcile unaligned views by learning latent common representations and employing a differentiable bi-graph matching algorithm for instance-level alignment [15]. Alternatively, MvCLN presented a category-level alignment paradigm, enhancing the efficiency and accuracy of cross-view alignment for partially view-unaligned clustering [39]. Subsequently, SURE further extended MvCLN to preserve comprehensive sample information by jointly reconstructing the original samples [38]. Despite their advancements, these methods predominantly focused on latent common representations, neglecting the specific information of each view, and thus failing to fully exploit the available comprehensive information [13, 41].

## 2.2 Contrastive Learning

Contrastive learning, renowned for its efficacy in representation learning tasks, has been widely integrated into numerous algorithms. SimCLR established a foundational framework by pairing two augmented views of the same sample as positive pairs, with other samples forming negative ones [5]. Although this cross-view contrastive learning achieved promising performance in various tasks, it risked introducing False Negative Pairs (FNPs), which share the same category while coming from different instances, within a batch in unsupervised settings, leading to performance degradation. Intuitively, addressing FNPs is crucial for boosting the robustness and effectiveness of contrastive learning. Various techniques have been proposed to tackle this issue: Yang et al. divide FNPs using a calculated threshold based on the distance between positives and negatives [38], while Huynh et al. employ a threshold and a top-k strategy to filter FNPs. Other approaches, like those by Chuang et al. [7] and Caron et al. [2], focus on increasing the weight of positive pairs to enhance the robustness. However, most of these methods roughly separate the training pairs into positives and negatives, which may not guarantee the correctness of the hard binary identification for all pairs, thereby limiting their performance improvement.

## 3 Methodology

In this section, we first formulate the Partially View-unaligned Problem (PVP) in subsection 3.1, and then provide an overview of VITAL in subsection 3.2. Subsequently, in the next two subsections, we elaborate on the two modules involved in VITAL, namely Variational Contrastive Learning and Dynamic Rectification.

## 3.1 Problem Formulation

Given a multi-view dataset $\mathcal{X} = \{X^1, X^2, \ldots, X^V\}$, where $V$ is the number of views, $X^k = \{x_1^k, x_2^k, \ldots, x_N^k\}$ represents the sample set of the $k$-th view, $N$ is the total number of instances, and $x_i^k$ denotes the $i$-th sample of the $k$-th view. Without loss of generality, considering the case of binary views, for a fully aligned dataset, we assume that $x_i^{k_1}$ and $x_i^{k_2}$ have alignment between $k1$-th view and $k2$-th view. However, the PVP arises when this alignment is disrupted, thus splitting the dataset into a fully aligned subset $A = \{a_i^1, a_i^2, \ldots, a_i^V\}_{i=1}^{N_1}$ and an unaligned subset $U = \{u_i^1, u_i^2, \ldots, u_i^V\}_{i=1}^{N_2}$.

## 3.2 Overview of VITAL

To address the mentioned PVP, existing methods primarily focus on seeking a latent common space for realignment by learning a mapping $f : \mathcal{X} \rightarrow \mathcal{Y}$ [15, 38, 39]. However, this often overlooks view-specific information from each view, consequently resulting in performance degradation. Drawing inspiration from the characteristics of Gaussian distribution, where the mean encapsulates the central feature and the variance reflects its diversity, we could model the shared information as the mean and the view-specific information as the variance. To achieve this, each sample is embedded as a Gaussian distribution instead of a fixed point in the latent space. More specifically, our VITAL employs variational inference to approximate the posterior Gaussian distribution of observed samples, minimizing the divergence from the true posterior through the proposed intra- and inter-view contrastive learning. Thus, the learned common representations (means) could facilitate category-level realignment due to its central semantics, while the view-specific representations (variances) provide complementary information due to the diverse specificity. The loss function of this methodology is formulated as:

$$\mathcal{L}_{VCL} = \mathcal{L}_{inter} + \mathcal{L}_{intra} + \mathcal{L}_{KL}. \tag{1}$$

Furthermore, to mitigate the adverse influence of FNPs, we propose a dynamic rectification module that integrates seamlessly with VITAL. This module employs a probabilistic model to infer a confidence mask for FNPs while dynamically correlating $\mathcal{L}_{inter}$ with this mask to establish a robust loss $\mathcal{L}_{inter-DR}$. The details of our VITAL will be elaborated in the following sections.

## 3.3 Variational Contrastive Learning

For an observed sample $x_i^k$, let $z_i^k$ represent its latent variables in the semantic space following [19]. We assume that $z_i^k$ adheres to a posterior Gaussian distribution $p_\Theta(z_i^k|x_i^k)$, where $\Theta$ indicates the true parameters. Due to sampling from $p_\Theta$ may yield latent variables semantically approximate to $z_i^k$, $p_\Theta$ encapsulates the essence of the observed sample. However, the sparsity of $z_i^k$ renders direct optimization of $p_\Theta(z_i^k|x_i^k)$ impractical [42]. To tackle this, variational inference is employed to approximate the distribution, thereby deriving intra- and inter- contrastive learning to enhance the discrimination of representations. More specifically, for each sample $x_i^k$, a recognition model $q_\phi(z_i^k|x_i^k) \sim \mathcal{N}(z_i^k; \mu_i^k, (\sigma_i^k)^2 I)$ serves as the approximation solution to $p_\Theta(z_i^k|x_i^k)$, where $\phi$ is the variational parameters [19]. Given that various views of an instance may serve

as mutual priors, the objective is to minimize the Kullback-Leibler (KL) divergence between the approximate and true posterior across all views:

$$\sum_{m=1}^{V}\sum_{n=1}^{V}D_{KL}(q_\phi(z_i|x_i^m)||p_\Theta(z_i|x_i^n))$$

$$= \sum_{m=1}^{V}\sum_{n=1}^{V}\int_z q_\phi(z_i|x_i^m)\log\frac{q_\phi(z_i|x_i^m)}{p_\Theta(z_i|x_i^n)}dz \qquad (2)$$

$$= \sum_{m=1}^{V}\sum_{n=1}^{V}\left[\log p_\Theta(x_i^n) - \mathcal{L}_{ELBO}\right],$$

where $\mathcal{L}_{ELBO}$ denotes the evidence lower bound (ELBO) [17]. Since $\log p_\Theta(x_i^n)$ remains constant for the observed sample, minimizing Equation (2) is equivalent to maximizing $\mathcal{L}_{ELBO}$:

$$\sum_{m=1}^{V}\sum_{n=1}^{V}\mathcal{L}_{ELBO} = -\sum_{m=1}^{V}\sum_{n=1}^{V}D_{KL}(q_\phi(z_i|x_i^m)||p_\Theta(z_i))$$

$$+ \sum_{m=1}^{V}\sum_{n=1}^{V}\mathbb{E}_{q_\phi(z_i|x_i^m)}[\log p_\Theta(x_i^n|z_i)], \qquad (3)$$

where $p_\Theta(z_i)$ could be a preset arbitrary distribution. Assuming that $p_\Theta(z_i)$ follows a standard Gaussian distribution $\mathcal{N}(z_i; \mathbf{0}, I)$, the first term in Equation (3) could be optimized via a KL loss:

$$\mathcal{L}_{KL} = \sum_{m=1}^{V}\sum_{n=1}^{V}D_{KL}(q_\phi(z_i|x_i^m)||p_\Theta(z_i))$$

$$= \sum_{m=1}^{V}\sum_{n=1}^{V}\frac{1}{2}(-\log(\sigma_i^m)^2 + (\mu_i^m)^2 + (\sigma_i^m)^2 - 1). \qquad (4)$$

Moreover, the second term in Equation (3) is used to measure the generative quality of the approximate posterior $q_\phi$, which can be calculated through intra- and inter-view contrastive losses:

$$-\sum_{m=1}^{V}\sum_{n=1}^{V}\mathbb{E}_{q_\phi(z_i|x_i^m)}[\log p_\Theta(x_i^n|z_i)]$$

$$= \underbrace{-\sum_{k=1}^{V}\mathbb{E}_{q_\phi(z_i^k|x_i^k)}[\log p_\Theta(x_i^k|z_i^k)]}_{\mathcal{L}'_{intra}} \underbrace{-\sum_{m\neq n}^{V}\mathbb{E}_{q_\phi(z_i^m|x_i^m)}[\log p_\Theta(x_i^n|z_i^n)]}_{\mathcal{L}'_{inter}},$$

$$(5)$$

where $\mathcal{L}'_{intra}$ and $\mathcal{L}'_{inter}$ are intra- and inter- contrastive losses respectively. By using the SGVB estimator [19], the two terms in Equation (5) can be approximated as a sampling process:

$$\mathcal{L}'_{intra} \simeq -\frac{1}{L}\sum_{l=1}^{L}\sum_{k=1}^{V}\log p_\Theta(x_i^k|z_{(i,l)}^k), \qquad (6)$$

where $z_{(i,l)}^k \sim q_\phi(z_i|x_i^k)$ and

$$\mathcal{L}'_{inter} \simeq -\frac{1}{L}\sum_{l=1}^{L}\sum_{m\neq n}^{V}\log p_\Theta(x_i^m|z_{(i,l)}^n), \qquad (7)$$

where $z_{(i,l)}^n \sim q_\phi(z_i|x_i^n)$, and $L$ is the number of Monte Carlo samples in the SGVB estimator, which can be set to 1 if the batch

size is large enough [19]. Accordingly, Equation (6) could be relaxed as follows:

$$\mathcal{L}_{intra} = \sum_{k=1}^{V}\mathcal{H}(T_{intra}, p(x_i^k, \hat{x}_i^k)), \qquad (8)$$

where $\mathcal{H}$ denotes cross-entropy, $T_{intra}$ represents the intra-view ground truth, $\hat{x}_i^k = q_\theta(z_{(i,l)}^k)$ is the result obtained through the reparameterization trick [19] via a generative model $q_\theta$, and $p(\cdot, \cdot)$ is the likelihood between two points, which is computed as follows:

$$p(b_i, b_j) = \frac{exp(sim(b_i, b_j)/\tau)}{\sum_{k=1}^{N}exp(sim(b_i, b_k)/\tau)}, \qquad (9)$$

where $sim(\cdot, \cdot)$ is the calculation function of cosine similarity, and $\tau$ denotes the temperature parameter.

For Equation (7), due to the inconsistency of the cross-view variables, direct computation of the expectation through sampling is not feasible. Since the specificity (variance) of each view is independent of each other, we relax the optimization of Equation (7) by maximizing the shared semantics across different views as follows:

$$\mathcal{L}_{inter} = \sum_{m\neq n}^{V}\mathcal{H}(T_{inter}, p(\mu_i^m, \mu_i^n)), \qquad (10)$$

where $T_{inter}$ is the inter-view ground truth. Therefore, the final variational contrastive learning loss function can be written as Equation (1). Notably, $T_{inter} = I$ may inadvertently introduce False Negative Pairs (FNPs) since $\mathcal{L}_{inter}$ is conducted at category level, potentially impending model efficacy [27]. The following section will introduce a Dynamic Rectification module designed to produce good soft rather than hard partitions of the pairs, thus enhancing the robustness.

### 3.4 Dynamic Rectification

In unsupervised contrastive learning, handling the challenge posed by FNPs is crucial for enhancing model performance. To this end, inspired by [45], we introduce a robust mechanism to improve the traditional contrastive learning by rewriting $T_{inter}$ as follows:

$$T_{inter} = p^\alpha I, \qquad (11)$$

where $\alpha$ is a parameter sensitive to FNPs. Equation (11) could produce gradients in different directions within two distinct intervals, whose dividing point can be computed by setting the gradient of $\mathcal{L}_{inter}$ with respect to $p$ to zero:

$$\Delta = \frac{\partial \mathcal{L}_{inter}}{\partial p} = -\alpha p^{\alpha-1}\log p - p^\alpha\frac{1}{p}$$

$$= -p^{\alpha-1}(1 + \alpha\log p) = 0, \qquad (12)$$

then $p = e^{-1/\alpha}$. As a result, the optimization surface is divided into two areas, i.e., $(0 < p < e^{-1/\alpha})$ and $(e^{-1/\alpha} < p < 1)$. In the two areas, the optimizer performs different optimization directions:

- $(0 < p < e^{-1/\alpha})$: Pairs in this interval are prone to be FNPs, mistakenly considered as hard positive samples due to low similarity by traditional contrastive learning. This misclassification will mislead the learning process, resulting in an overfitting issue [12, 45]. In contrast, our $\mathcal{L}_{inter}$ applies a reverse gradient to these pairs, facilitating their correct optimization.

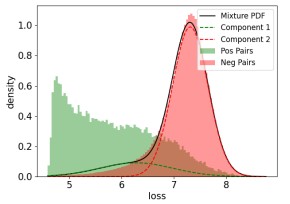 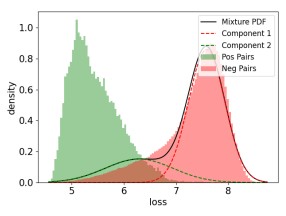 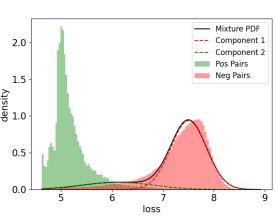 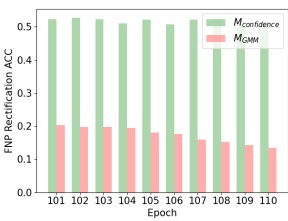

(a) Loss after $\mathcal{L}_{VCL}$   (b) Loss with $M_{GMM}$   (c) Loss with $M_{confidence}$   (d) Rectification Accuracy

**Figure 2: Visualization of different masking schemes on the Deep Caltech-101 dataset in partially aligned (50%) scenario. We present the density plots of loss at different training stages in (a), (b), and (c). (a) After the initial training stage with $\mathcal{L}_{VCL}$, there still exists considerable overlap between positives and negatives. (b) Directly employing $M_{GMM}$ leads to suboptimal results, with partial overlap remaining between positives and negatives. (c) By utilizing $M_{confidence}$, which discards the pairs with confounding in $M_{GMM}$, this overlap can be remarkably reduced. (d) Compared to $M_{GMM}$, $M_{confidence}$ exhibits higher FNP rectification accuracy.**

- $(e^{-1/\alpha} < p < 1)$: Pairs within this range are likely true positive pairs, thus receiving the correct optimization direction. Before dynamically rectifying FNPs, we initially set $\alpha$ to 0.1 to mitigate the impact of FNPs during the rectification process and guarantee correct optimization.

After the aforementioned steps, the next challenge is to rectify the FNPs caused by category-level mismatching. In the past, probabilistic models have been proven to be useful for decoupling loss distributions between noisy labels and clean labels [1, 16, 23]. Inspired by this, we similarly propose to fit the loss distribution of all pairs using a two-component Gaussian Mixture Model (GMM):

$$p(l|\theta) = \sum_{k=1}^{2} \gamma_k \phi(l|k), \qquad (13)$$

where $l$ represents the contrastive loss for a specific pair in order to ensure a fair comparison, $\gamma_k$ and $\phi(l|k)$ are the mixture coefficient and the probability density of the $k$-th component [16] respectively, which can be simply solved using the EM algorithm [9]. Moreover, since the loss of positives is small, we consider the component with a lower mean as the distribution of positives, while the other as negatives. After that, we can easily compute the posterior probability of all pairs with respect to a specific component as follows:

$$w_i = p(k|l_i) = \frac{p(k)p(l_i|k)}{p(l_i)}. \qquad (14)$$

If $k$ represents the component with a lower mean, then $\{w_i\}_{i=1}^{B^2}$ can be regarded as the probability that all pairs belong to positives, or alternatively, as soft labels for FNPs. As positives should have smaller losses, we dynamically associate $\{w_i\}_{i=1}^{B^2}$ with the sensitivity parameter in $T_{inter}$ through $\alpha' = 0.1 \times \frac{w_i - \min(\{w_i\}_{i=1}^{B^2})}{\max(\{w_i\}_{i=1}^{B^2}) - \min(\{w_i\}_{i=1}^{B^2})}$, where $B$ represents the batch size. This implies utilizing the posterior probabilities provided by the GMM model to guide the optimization of $\mathcal{L}_{inter}$.

On the other hand, we can easily separate the positive set and negative set by setting a threshold of $\{w_i\}_{i=1}^{B^2}$ [16]. However, in practical applications, we found that directly using the GMM model for predictions in the overlapping region between two components

is often inaccurate, as illustrated in Figure 2 (a) and (b). To address this issue, we propose a solution called confidence mask to dynamically focus more on clean pairs and less on noisy ones. Specifically, for the fitting result $\phi(l|k) \sim \mathcal{N}(\mu_k, \sigma_k^2 I)$, we first calculate the confidence through the distance between the means of the two components:

$$d = \left| \frac{\mu_1 - \mu_2}{\max(\{l_i\}_{i=1}^{B^2}) - \min(\{l_i\}_{i=1}^{B^2})} \right|. \qquad (15)$$

Then, for the initial version of the positive set $M_{GMM}$ obtained by dividing $\{w_i\}_{i=1}^{B^2}$ with a threshold of 0.5 [16], we pick out the subset with higher confidence by

$$M_{confidence} = \{l \in M_{GMM} | l \leq Q_d\}, \qquad (16)$$

where $Q_d$ represents the $d$ quantile of the loss value set $\{l_i\}_{i=1}^{B^2}$ and we can finally take $M_{confidence}$ as the final positive set. This approach avoids pairs in the overlapping part between the two components and improves the rectification accuracy for FNPs. Figure 2 (c) and (d) demonstrate the effectiveness of $M_{confidence}$ through experiments. Therefore, the dynamically robust version of $\mathcal{L}_{inter}$ can be written as:

$$\mathcal{L}_{inter-DR} = \sum_{m \neq n}^{V} \mathcal{H}(T'_{inter}, p(\mu_i^m, \mu_i^n)), \qquad (17)$$

where $T'_{inter} = p^{\alpha'} \times M$, $M$ represents the mask of the final positive set, and $\times$ denotes element-wise multiplication. In summary, the final loss function of this module can be rewritten as:

$$\mathcal{L}_{VCL-DR} = \mathcal{L}_{inter-DR} + \mathcal{L}_{intra} + \mathcal{L}_{KL}. \qquad (18)$$

Through Equation (18), we integrate the process of dynamically rectifying FNPs with our robust contrastive loss. In the next section, we will demonstrate the specific implementation details of applying this framework to PVP.

### 3.5 Implementation Details

Given an observed set $X$, we first use $\mathcal{L}_{VCL}$ to perform the first stage training. Subsequently, we replace $\mathcal{L}_{VCL}$ with $\mathcal{L}_{VCL-DR}$ to rectify the FNPs, thereby securing a more robust model. The mean

**Table 1: The partially aligned (50%) clustering performance on eight widely used multi-view datasets. The best results are indicated in bold, and the second-best results are indicated with an underline.**

| Methods | CUB | | | Scene-15 | | | WIKI | | | NUS-WIDE | | |
|---|---|---|---|---|---|---|---|---|---|---|---|---|
| | ACC | NMI | ARI | ACC | NMI | ARI | ACC | NMI | ARI | ACC | NMI | ARI |
| AE2-Nets (CVPR'19) | 31.32 | 25.55 | 12.62 | 29.90 | 29.49 | 14.45 | 17.66 | 4.32 | 1.89 | 14.61 | 3.91 | 2.36 |
| PVC (NeurIPS'20) | 55.53 | 59.75 | 44.47 | 37.88 | 39.12 | 20.63 | 27.21 | 17.59 | 8.94 | 48.45 | 42.39 | 33.10 |
| MvCLN (CVPR'21) | 52.13 | 49.67 | 35.54 | 38.53 | 39.90 | 24.26 | 36.00 | 17.52 | 13.11 | 57.33 | 42.40 | 36.22 |
| DSIMVC (ICML'22) | 35.87 | 31.70 | 16.71 | 29.65 | 31.20 | 14.91 | 17.11 | 3.59 | 1.57 | 35.30 | 25.29 | 16.24 |
| MFLVC (CVPR'22) | 40.97 | 38.67 | 22.61 | 31.83 | 34.10 | 17.16 | 17.93 | 3.81 | 1.76 | 24.24 | 15.01 | 9.10 |
| DCP (TPAMI'23) | 31.07 | 28.76 | 6.65 | 29.48 | 30.70 | 12.85 | 17.54 | 3.17 | 1.09 | 32.45 | 22.96 | 13.39 |
| GCFAgg (CVPR'23) | 39.87 | 39.36 | 22.24 | 31.26 | 34.83 | 17.08 | 17.42 | 3.79 | 1.71 | 26.87 | 16.96 | 9.28 |
| DealMVC (MM'23) | 40.87 | 37.55 | 22.18 | 32.79 | 34.08 | 17.56 | 17.93 | 3.97 | 1.83 | 32.02 | 20.11 | 12.49 |
| SURE (TPAMI'23) | 54.47 | 50.26 | 37.19 | 40.32 | 40.33 | 23.08 | 34.28 | 16.90 | 11.69 | 56.69 | 42.66 | 36.76 |
| ICMVC (AAAI'24) | 58.73 | 53.15 | 40.11 | 32.45 | 33.74 | 18.12 | 17.26 | 3.75 | 1.68 | 45.36 | 29.06 | 22.88 |
| **VITAL** | **78.70** | **75.74** | **65.40** | **41.05** | **41.76** | **24.93** | **36.57** | **20.58** | **15.07** | **62.91** | **47.86** | **42.74** |

| Methods | Deep Animal | | | Deep Caltech-101 | | | MNIST-USPS | | | NoisyMNIST | | |
|---|---|---|---|---|---|---|---|---|---|---|---|---|
| | ACC | NMI | ARI | ACC | NMI | ARI | ACC | NMI | ARI | ACC | NMI | ARI |
| AE2-Nets (CVPR'19) | 6.43 | 7.64 | 0.62 | 7.67 | 14.22 | 1.78 | 41.15 | 38.13 | 24.12 | 32.53 | 26.29 | 16.06 |
| PVC (NeurIPS'20) | 3.83 | 0.00 | 0.00 | 18.59 | 48.89 | 14.60 | 86.54 | 78.08 | 74.60 | 81.84 | 82.29 | 82.03 |
| MvCLN (CVPR'21) | 26.24 | 40.24 | 19.74 | 35.55 | 60.99 | 40.90 | 89.96 | 81.36 | 80.40 | 91.05 | 84.15 | 83.56 |
| DSIMVC (ICML'22) | 14.60 | 22.20 | 5.35 | 17.07 | 25.54 | 6.84 | 34.52 | 30.03 | 18.24 | 24.23 | 14.62 | 8.28 |
| MFLVC (CVPR'22) | 12.82 | 17.89 | 4.56 | 19.49 | 31.34 | 9.74 | 33.88 | 29.66 | 18.04 | 22.96 | 14.38 | 7.74 |
| DCP (TPAMI'23) | 12.06 | 17.52 | 4.44 | 16.33 | 38.08 | 11.76 | 31.84 | 26.16 | 7.02 | 23.47 | 15.13 | 3.70 |
| GCFAgg (CVPR'23) | 12.44 | 19.20 | 4.39 | 21.33 | 49.22 | 18.82 | 36.80 | 29.92 | 18.09 | 23.11 | 13.52 | 6.67 |
| DealMVC (MM'23) | 14.73 | 20.15 | 5.47 | 22.05 | 29.02 | 11.18 | 32.78 | 25.22 | 14.95 | 25.60 | 16.09 | 9.34 |
| SURE (TPAMI'23) | 27.65 | 40.76 | 19.85 | 46.18 | 70.68 | 32.98 | 92.14 | 82.83 | 83.47 | 95.17 | 88.24 | 89.72 |
| ICMVC (AAAI'24) | 12.33 | 15.30 | 4.41 | 23.30 | 37.58 | 18.31 | 34.95 | 29.18 | 18.18 | 27.23 | 18.99 | 10.59 |
| **VITAL** | **44.51** | **49.04** | **29.87** | **53.97** | **74.05** | **52.40** | **94.17** | **85.91** | **87.52** | **95.44** | **88.66** | **90.25** |

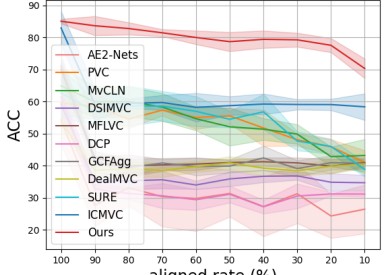
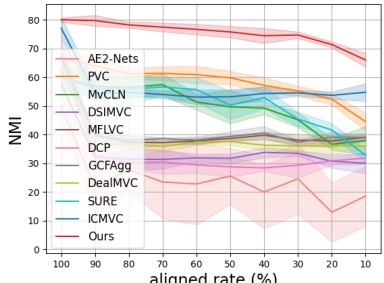
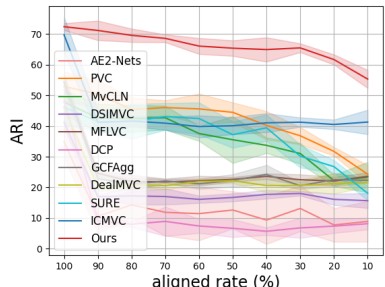

**Figure 3: The clustering performance of the CUB dataset under different aligned rates. The solid lines represent the average results of five different random seeds, while the shaded regions indicate the standard deviation range of the results.**

$\mu_i^k$ and standard deviation $\sigma_i^k$ of each sample are utilized as its common and specific representations, respectively:

$$
\begin{aligned}
C^k &= \left[\mu_1^k, \mu_2^k, \ldots, \mu_N^k\right]^T \\
\Sigma^k &= \left[\sigma_1^k, \sigma_2^k, \ldots, \sigma_N^k\right]^T.
\end{aligned}
\tag{19}
$$

Finally, $\{C^k\}_{k=1}^V$ facilitates category-level realignment across different views, while the fused representations $\{z_i\}_{i=1}^N$ are used for clustering, where $z_i = [\mu_i^k + \sigma_i^k]_{k=1}^V$. The fused representations incorporate both common and specific view information, offering comprehensiveness and interpretability. Classical clustering

algorithms, such as K-means, can then be applied to the fused representations to obtain the final clustering results.

## 4 Experiment

In this section, we validate the effectiveness of VITAL by evaluating the clustering performance on eight widely used multi-view datasets in both partially and fully aligned scenarios. Firstly, we introduce the experimental setups in Section 4.1, including datasets and training settings. Subsequently, in Section 4.2, we compare the clustering performance of VITAL with ten state-of-the-art deep clustering methods. After that, Section 4.3 demonstrates the effectiveness of the Dynamic Rectification module in VITAL through

**Table 2: The fully aligned (100%) clustering performance on eight widely used multi-view datasets. The best results are indicated in bold, and the second-best results are indicated with an underline.**

| Methods | CUB | | | Scene-15 | | | WIKI | | | NUS-WIDE | | |
|---|---|---|---|---|---|---|---|---|---|---|---|---|
| | ACC | NMI | ARI | ACC | NMI | ARI | ACC | NMI | ARI | ACC | NMI | ARI |
| AE2-Nets (CVPR'19) | 54.33 | 49.93 | 34.88 | 37.17 | 40.47 | 22.24 | 48.69 | 44.25 | 33.01 | 13.88 | 1.88 | 1.02 |
| PVC (NeurIPS'20) | 59.57 | 66.69 | 52.90 | 38.01 | 39.82 | 21.06 | 39.40 | 42.74 | 28.42 | 54.14 | 46.42 | 38.78 |
| MvCLN (CVPR'21) | 64.92 | 59.96 | 47.84 | 37.90 | 42.31 | 25.58 | 50.50 | 45.46 | 32.73 | 58.70 | 46.00 | 39.59 |
| DSIMVC (ICML'22) | 59.13 | 57.50 | 41.20 | 30.69 | 35.17 | 17.06 | 43.80 | 34.45 | 26.63 | 55.61 | 46.16 | 37.42 |
| MFLVC (CVPR'22) | 70.03 | 67.04 | 54.44 | 37.26 | 40.58 | 21.89 | 44.56 | 38.47 | 26.91 | 56.89 | 46.15 | 38.15 |
| DCP (TPAMI'23) | 61.53 | 68.27 | 48.47 | 39.41 | 41.47 | 21.00 | 46.12 | 43.69 | 26.91 | 51.99 | 42.65 | 28.67 |
| GCFAgg (CVPR'23) | 71.17 | 67.12 | 54.35 | 37.63 | 40.39 | 21.96 | 52.80 | 48.21 | 35.83 | 51.31 | 40.93 | 31.03 |
| DealMVC (MM'23) | 53.66 | 62.35 | 46.07 | 39.66 | 42.38 | 24.86 | **56.89** | _51.36_ | **42.78** | 55.28 | 41.80 | 34.87 |
| SURE (TPAMI'23) | 62.70 | 60.06 | 46.13 | _42.75_ | 42.48 | 24.57 | 53.25 | 46.64 | 35.21 | 58.14 | 46.06 | 39.48 |
| ICMVC (AAAI'24) | _82.97_ | _77.05_ | _69.75_ | 41.18 | _43.62_ | _25.73_ | 52.99 | 44.42 | 35.91 | _66.38_ | _52.73_ | _47.67_ |
| **VITAL** | **85.07** | **79.99** | **72.40** | **42.84** | **46.24** | **28.01** | _54.16_ | **53.29** | _41.38_ | **66.72** | **54.42** | **49.21** |

| Methods | Deep Animal | | | Deep Caltech-101 | | | MNIST-USPS | | | NoisyMNIST | | |
|---|---|---|---|---|---|---|---|---|---|---|---|---|
| | ACC | NMI | ARI | ACC | NMI | ARI | ACC | NMI | ARI | ACC | NMI | ARI |
| AE2-Nets (CVPR'19) | 10.02 | 18.99 | 2.44 | 7.59 | 12.48 | 0.81 | 83.08 | 80.73 | 74.73 | 42.11 | 43.38 | 30.42 |
| PVC (NeurIPS'20) | 3.83 | 0.00 | 0.00 | 20.54 | 51.40 | 15.66 | 95.28 | 90.36 | 90.05 | 87.10 | 92.84 | 93.14 |
| MvCLN (CVPR'21) | 35.28 | _54.19_ | 29.37 | 39.55 | 65.29 | 32.81 | 98.76 | 96.47 | 97.27 | 97.30 | 94.16 | 95.31 |
| DSIMVC (ICML'22) | 27.12 | 41.73 | 16.43 | 21.38 | 37.01 | 17.16 | 99.37 | 98.23 | 98.60 | 57.07 | 54.88 | 43.61 |
| MFLVC (CVPR'22) | 26.92 | 39.99 | 17.94 | 49.43 | 73.39 | 34.33 | _99.53_ | _98.64_ | _98.95_ | 97.56 | 93.74 | 94.75 |
| DCP (TPAMI'23) | 27.81 | 45.44 | 18.33 | 49.16 | 74.41 | _48.98_ | 99.11 | 97.50 | 98.04 | 81.19 | 86.01 | 75.76 |
| GCFAgg (CVPR'23) | 30.79 | 43.85 | 21.44 | _55.65_ | _80.67_ | 37.49 | 98.37 | 96.19 | 96.44 | 88.58 | 86.55 | 80.68 |
| DealMVC (MM'23) | _36.07_ | 49.34 | 26.36 | 17.38 | 23.13 | 10.24 | 93.07 | 95.30 | 91.94 | _98.64_ | **96.70** | _97.15_ |
| SURE (TPAMI'23) | 35.76 | 53.62 | _29.51_ | 43.77 | 70.05 | 29.46 | 99.12 | 97.49 | 98.05 | 98.39 | 95.41 | 96.50 |
| ICMVC (AAAI'24) | 24.55 | 49.94 | 21.05 | 34.02 | 64.02 | 39.97 | 99.33 | 98.08 | 98.52 | **98.72** | _96.25_ | **97.19** |
| **VITAL** | **55.59** | **64.74** | **44.62** | **65.02** | **82.02** | **61.52** | **99.78** | **99.36** | **99.52** | 98.36 | 95.38 | 96.42 |

ablation studies. Finally, in Section 4.4, we present the visualization results.

## 4.1 Experimental Setups

Eight widely used multi-view datasets were utilized in our experiments, including CUB [33], Scene15 [11], WIKI [8], NUS-WIDE [6], Deep Animal [21], Deep Caltech-101 [10], MNIST-USPS [28], and NoisyMNIST [35]. For the partially aligned scenario, we simulate it by randomly shuffling the correspondence of a portion of the original dataset. The aligned rate is defined as $\eta = \frac{N_1}{N}$, where $N_1$ represents the number of samples with correct alignments, and $N$ is the total number of instances.

To train VITAL, we used PyTorch version 1.12.1 with an NVIDIA 3090Ti for all experiments. Our training strategy consists of two stages: initially, we perform variational contrastive learning using $\mathcal{L}_{VCL}$ to obtain the approximate posterior distribution of samples, for a total of 100 epochs. Subsequently, we replace $\mathcal{L}_{VCL}$ with $\mathcal{L}_{VCL-DR}$ for dynamic rectification of FNPs, conducting additional 10 epochs to enhance model performance. As for the hyperparameters, We employed the Adam [18] optimizer with a learning rate of 2e-3 for the first stage and 1e-4 for the second, and we fixed the contrastive temperature $\tau$ to 0.4 for training on all datasets. Regarding the network architecture, we adopted the same four-layer fully connected network for the probabilistic encoder/decoder across all datasets. It is worth noting that in VITAL, all datasets share the same model structure, which allows it to be seamlessly applied to

various PVP downstream task scenarios but not limited to the field of clustering.

## 4.2 Comparisons with State-Of-The-Art

In this section, we conducted a comparison between VITAL and ten state-of-the-art deep clustering methods in both partially and fully aligned scenarios. The methods include AE2-Nets [44], PVC [15], MvCLN [39], DSIMVC [30], MFLVC [36], DCP [25], GCFAgg[37], DealMVC[40], SURE [38] and ICMVC [3]. As only PVC, MvCLN, and SURE can directly handle partially aligned scenarios, before applying other algorithms for comparison, we first reduce the dimensionality of the original features using PCA and then realign the representations via the Hungarian algorithm [20]. After that, we run all methods five times and report the average performance in terms of three widely used metrics, including clustering Accuracy (ACC), Normalized Mutual Information (NMI), and Adjusted Rand Index (ARI), where larger values indicate better clustering performance. Table 1 and Table 2 show the performance in partially and fully aligned scenarios, respectively. Figure 3 displays the performance curves of each method under different aligned rates from 10% to 100%. From these experimental results, we can easily observe the following:

- In the partially aligned scenario, VITAL achieves state-of-the-art performance on all datasets, with significant gaps observed between it and other baselines on CUB, NUS-WIDE, Deep Animal, and Deep Caltech-101. This verifies the importance of leveraging specific information for complementary

**Table 3: Ablation studies of our method. ✔ denotes the adoption of the component, while ✗ indicates its exclusion. The best results are indicated in bold.**

| Aligned | $\mathcal{L}_{VCL-DR}$ | CUB | | | Scene15 | | | WIKI | | | NUS-WIDE | | |
|---|---|---|---|---|---|---|---|---|---|---|---|---|---|
| | | ACC | NMI | ARI | ACC | NMI | ARI | ACC | NMI | ARI | ACC | NMI | ARI |
| Partially | ✗ | 77.83 | 74.40 | 64.25 | 39.55 | 38.55 | 22.54 | 35.59 | 19.91 | 14.37 | 59.77 | 45.58 | 38.60 |
| | ✔ | **78.70** | **75.74** | **65.40** | **41.05** | **41.76** | **24.93** | **36.57** | **20.58** | **15.07** | **62.91** | **47.86** | **42.74** |
| Fully | ✗ | 84.70 | 79.34 | 71.72 | 42.26 | 43.88 | 26.10 | 53.45 | 52.77 | 40.47 | 63.04 | 51.42 | 43.90 |
| | ✔ | **85.07** | **79.99** | **72.40** | **42.84** | **46.24** | **28.01** | **54.16** | **53.29** | **41.38** | **66.72** | **54.42** | **49.21** |
| Aligned | $\mathcal{L}_{VCL-DR}$ | Deep Animal | | | Deep Caltech-101 | | | MNIST-USPS | | | NoisyMNIST | | |
| | | ACC | NMI | ARI | ACC | NMI | ARI | ACC | NMI | ARI | ACC | NMI | ARI |
| Partially | ✗ | 42.99 | 48.01 | 28.42 | 45.09 | 70.17 | 31.62 | 92.56 | 82.55 | 84.21 | 94.21 | 87.22 | 87.93 |
| | ✔ | **44.51** | **49.04** | **29.87** | **53.97** | **74.05** | **52.40** | **94.17** | **85.91** | **87.52** | **95.44** | **88.66** | **90.25** |
| Fully | ✗ | 54.02 | 62.58 | 42.28 | 54.06 | 78.46 | 36.84 | 99.75 | 99.25 | 99.44 | 92.34 | 92.01 | 88.36 |
| | ✔ | **55.59** | **64.74** | **44.62** | **65.02** | **82.02** | **61.52** | **99.78** | **99.36** | **99.52** | **98.36** | **95.38** | **96.42** |

sample information, as the selected baselines do not explicitly consider it.

- For PVP tasks, the comprehensiveness of the view representations learned by our framework ensures consistent superiority across different aligned rates. This advantage persists even under extreme scenarios (10% aligned).
- Superior performance are obtained by VITAL in both partially and fully aligned scenarios, indicating its versatility beyond being a specific method tailored solely for PVP. We attribute this success to the contributions of both common and specific information, as well as the effective rectification of FNPs in unsupervised contrastive learning.

## 4.3 Ablation Studies

In this section, we conduct ablation studies on the Dynamic Rectification module in VITAL. Specifically, for all eight datasets used in the experiments, we analyze the impact of the $\mathcal{L}_{VCL-DR}$ loss term on model performance in both partially and fully aligned scenarios. The clustering performance are shown in Table 3, where it is evident that when $\mathcal{L}_{VCL-DR}$ is not used to rectify False Negative Pairs (FNPs) in the second dynamic training stage, there is a noticeable decline in performance across all datasets to varying degrees. This verifies the effectiveness of $\mathcal{L}_{VCL-DR}$.

## 4.4 t-SNE visualization

In Figure 4, we present the t-SNE [32] visualization results of SURE [38] (the state-of-the-art method for addressing PVP) and our VITAL on the MNIST-USPS dataset. It is evident that VITAL exhibits larger clusters and fewer outliers compared to SURE. We attribute this to VITAL's consideration of both common and specific information of the samples. In contrast to SURE, which only utilizes common information for clustering, VITAL can achieve higher clustering accuracy.

## 5 Conclusions

In this paper, we propose a framework explicitly considering the common and specific information across different views to address

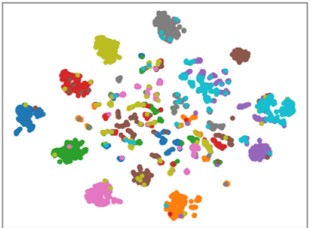 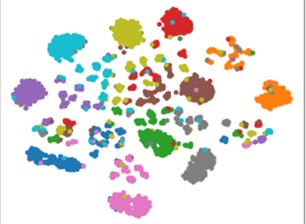

**(a) SURE (ACC=92.98)**     **(b) VITAL (ACC=94.80)**

**Figure 4: The t-SNE visualization results of the MNIST-USPS dataset in partially aligned (50%) scenario, where data are colored by classes.**

the Partially View-unaligned Problem (PVP). Diverging from existing works, our contributions are primarily twofold: i) a novel variational contrastive learning paradigm is proposed to learn the common and specific information from multi-view data. ii) to tackle the challenge of False Negative Pairs (FNPs) in unsupervised contrastive learning, we present a robust loss that dynamically associates the rectification process of FNPs with contrastive learning. We demonstrate the effectiveness of our framework in multi-view clustering in both partially and fully aligned scenarios. In future research, we aim to refine our method to handle cases involving Partially Sample-missing Problem (PSP) and combinations such as PSP+PVP.

## Acknowledgments

This work was supported by the National Natural Science Foundation of China under Grants U21B2040, 62176171, 62102274; by the Sichuan Science and Technology Program under Grant 2024NS-FTD0038; by the Fundamental Research Funds for the Central Universities under Grants CJ202303 and CJ202403; by the System of Systems and Artificial Intelligence Laboratory pioneer fund grant; by the EDB Space Technology Development Grant under Project S22-19016-STDP.

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
