# OpenReview forum: "Robust Variational Contrastive Learning for Partially View-unaligned Clustering"
_acmmm.org/ACMMM/2024/Conference — MM2024 Poster_

### Official Review · Reviewer_zLcq · 2024-05-15

**Rating:** 6
**Confidence:** 4

**Summary:**

This paper proposes a robust framework called VITAL for partially view-unaligned clustering. VITAL learns both common and specific information simultaneously by modeling each data sample as a Gaussian distribution in the latent space and employing variational inference and contrastive learning. The framework also incorporates a dynamic rectification module to handle false negative pairs generated during unsupervised contrastive learning.

**Strengths:**

1. The paper addresses the problem of partially view-unaligned clustering, which is more realistic than the fully aligned assumption in many real-world scenarios.
2. The proposed VITAL framework is novel in its approach to learning both common and specific information simultaneously by modeling data samples as Gaussian distributions in the latent space.
3. The use of variational inference and contrastive learning to preserve comprehensive semantic information is a theoretically sound approach.
4. The introduction of a robust contrastive loss to handle false negative pairs is an innovative solution to a common problem in unsupervised contrastive learning.
5. The experimental evaluation is adequate, with comparisons to ten state-of-the-art deep clustering methods on eight benchmark datasets, covering both partially and fully aligned scenarios.
6. The paper is well-written and clearly organized, making it easy to follow the proposed methodology and results.

**Limitations:**

1. While the paper mentions that the proposed VITAL framework outperforms state-of-the-art methods, it would be beneficial to provide more detailed comparisons and analyses of the performance gaps, especially on the CUB, NUS-WIDE, Deep Animal, and Deep Caltech-101 datasets.
2. The paper does not provide a thorough discussion on the computational complexity of the proposed framework, which could be an important factor in real-world applications.
3. The paper lacks a sensitivity analysis of the hyperparameters involved in the VITAL framework, which could help practitioners better understand the robustness and generalizability of the proposed method.
4. Although the paper addresses the partially view-unaligned problem, it does not discuss how the proposed framework would perform in scenarios with extreme view misalignment or missing views.

**Suitability:**

3

---

### Official Review · Reviewer_hk4X · 2024-05-15

**Rating:** 5
**Confidence:** 3

**Summary:**

This paper presents a robust framework for partially view-unaligned clustering, named VITAL. It addresses the challenges of incomplete and unaligned views by learning both common and specific information simultaneously. VITAL uses variational contrastive learning to model each data sample as a Gaussian distribution in the latent space, capturing both shared and view-specific semantics. The framework also includes a dynamic rectification module to handle false negative pairs generated during unsupervised contrastive learning.

**Strengths:**

- Novelty: This paper introduce variational inference to model both common and specific information across different views, which is a novel idea.
- Technical correctness: The proposed method is based on solid theoretical foundations, including variational inference and contrastive learning.
- Adequate evaluation: The authors perform extensive experiments on eight benchmark datasets in both partially aligned and fully aligned scenarios, comparing their method with ten state-of-the-art baselines. The results show consistent improvements over existing methods, demonstrating the effectiveness of VITAL.

**Limitations:**

- This paper not provide a clear ablation study to demonstrate the individual contributions of the contrastive learning and dynamic rectification modules to the overall performance of the VITAL framework.
- This paper lacks a discussion on the scalability of the proposed method to larger datasets or higher-dimensional data, which could be important for real world applications.
- This paper does provide a detailed analysis of the learned Gaussian distributions in the latent, which could offer insights into the effectiveness of the proposed in capturing common and specific information.
- This paper lacks the parameter analysis about 𝛼, which is critical to observe the influence of FNPs.

**Suitability:**

3

---

### Official Review · Reviewer_DcQS · 2024-05-24

**Rating:** 4
**Confidence:** 3

**Summary:**

The paper proposes a contrastive learning method based on variational autoencoder, named VITAL, to solve the problem of Partially View-unaligned Problem (PVP). By integrating common information and specific information to achieve comprehensive perception, and combining confidence masks to alleviate the negative impact of FNP during the training process.

**Strengths:**

1.	The Idea is simple and novel, and the motivation is clear.
2.	The article is well-organized and the sections are arranged reasonably.
3.	Good experimental results.

**Limitations:**

1.	ELBO as the evidence lower bound is supposed to maximize. It is necessary to adjust the position of the numerator and denominator inside the log to change the addition of LELBO to the subtraction of LELBO.
2.	Equations 6 and 7 represent reconstruction and consistency semantics, respectively. But I'm not quite sure how Equations 8 and 10 are derived based on Equations 6 and 7. It feels more like a direct implementation based on the semantics expressed by Equations 6 and 7 is provided here. Why adopt this approach? There are various reconstruction and consistency losses that exist.
3.	For the derivation formula given in the appendix. I think the first term of the fifth "equal sign" in Equation 1 should be multiplied by the coefficient V. The second item of the seventh "equal sign" should also be like this.
4.	In Equations 8 and 10, The p-function returns a number that does not match the dimensions of Tintra and Tinter. Therefore, this representation is not appropriate.

**Suitability:**

3

---

### Official Review · Reviewer_Vwi3 · 2024-06-10

**Rating:** 4
**Confidence:** 3

**Summary:**

The paper proposes a robust framework called VariatIonal ConTrAstive Learning (VITAL) to address the Partially View-unaligned Problem (PVP) in multi-view learning. VITAL models each data sample as a Gaussian distribution in the latent space, where the mean represents the common information across views, and the variance captures the view-specific information. To mitigate the negative impact of False Negative Pairs (FNPs) in unsupervised contrastive learning, VITAL introduces a robust contrastive loss that adaptively focuses more on clean pairs and less on noisy ones.

**Strengths:**

1. The method has a certain degree of innovation.
2. It achieves significant performance improvements.

**Limitations:**

1. There is a lack of explanation regarding the rationale behind the design. For example, why is it modeled as a Gaussian distribution? Why is variational inference introduced?
2. In Section 2.2, you point out the drawbacks of previous methods for solving the FNPs problem. What are the advantages of your method? Why does it avoid the mentioned drawbacks?
3. How does the performance vary with different missing rates?
4. Can you provide more analysis on why the proposed method is robust?

**Suitability:**

3

---

### Meta-Review · Area_Chair_42Se · 2024-07-02

**Recommendation:** Accept (Poster)
**Confidence:** 5

**Metareview:**

The initial ratings are all positive, and two reviewers raise their ratings after rebuttal. Since all ratings after rebuttal are above weak accept, this paper is accepted.